# Contextual factors influencing a training intervention aimed at improved maternal and newborn healthcare in a health zone of the Democratic Republic of Congo

**Malin Bogren**[1]*, **Sylvie Nabintu Mwambali**[2], **Marie Berg**[1,2]

**1** Institute of Health and Care Sciences, Sahlgrenska Academy, University of Gothenburg, Gothenburg, Sweden, **2** Faculty of Medicine and Community Health, Department of Obstetrics and Gynecology, Evangelical University of Africa, Bukavu, Democratic Republic of Congo

* malin.bogren@gu.se

**Data Availability Statement:** Data cannot be shared publicly because to do so could potentially

## Abstract

### Background

Maternal and neonatal mortality and morbidity in the Democratic Republic of Congo (DRC) are among the highest worldwide. As part of a quality improvement programme in a health zone in the DRC aimed at contributing to reduced maternal and neonatal mortality and morbidity, a three-pillar training intervention around childbirth was developed and implemented in collaboration between Swedish and Congolese researchers and healthcare professionals. The aim of this study is to explore contextual factors influencing this intervention.

### Methods

A qualitative research design was used, with data collected through focus group discussions (n = 7) with healthcare professionals involved in the intervention before and at the end (n = 9). Transcribed discussions were inductively analysed using content analysis.

### Results

Three generic categories describe the contextual factors influencing the intervention: i) Incentives motivated participants' efforts to begin a training programme; ii) Involving the local health authorities was important; and (iii) Having physical space, electricity, and equipment in place was crucial.

### Conclusions

This study and similar ones highlight that incentives of various types are crucial contextual factors that influence training interventions, and have to be considered already in the planning of such interventions. One such factor is expectations of monetary incentives. To meet this in a small research project like ours would require a reduction of the scale and thus limit the implementation of new evidence-based knowledge into practice aimed at reducing maternal mortality and morbidity.

identify our study participants and individual healthcare facilities, and assurances were given to respondents that any publication would not do so. Requests for access to the data underlying our findings will be considered by the National Ethical Committee of Public Health/CNES South Kivu province, and should be addressed to Prof Kitoka Moke at kitoka02@yahoo.fr.

**Funding:** The study was conducted with financial assistance from the Laerdal Foundation and Sahlgrensringen. The funders had no role in study design, data collection and analysis, decision to publish, or preparation of the manuscript.

**Competing interests:** The authors have declared no competing interests exist.

**Abbreviations:** HBB, Helping Babies Breathe; HMS, Helping Mothers Survive; DRC, Democratic Republic of Congo; FGDs, focus group discussions; SDG, Sustainable Development Goal.

## Introduction

In the Democratic Republic of Congo (DRC), the maternal and neonatal mortality ratio (MMR) remains high. According to the latest nationally reported statistics, in 2013–14 the maternal mortality rate was 846 deaths per 100,000 live births and the neonatal mortality rate 28 deaths per 1,000 live births [1]. This, despite the fact that 80% of births were assisted in healthcare facilities by skilled healthcare providers, consisting of either midwives, physicians, or nurses [2].

The provision of high-quality care is central in achieving health-related targets within the Sustainable Development Goals (SDGs), especially the health of mothers and newborns [3,4]. In the strive for high-quality care around childbirth, health systems need to ensure that all women and their newborns receive quality care, defined as being scientifically evidence-based, equitable, respectful, effective, timely, efficient, and person-centred [2,5–8]. Furthermore, health systems need to be adapted to context, i.e. the environment and surroundings in which a proposed change is to be implemented [9]. Poor quality of care is a greater barrier than insufficient access to healthcare in the DRC as well as worldwide [3,10].

The training of healthcare providers is known to be a useful tool for reducing maternal and newborn mortality; however, no specific training strategy is effective in every context [11], which means that the same interventions have different effects in different contexts. Context includes anything internal and external to an intervention that may act as a barrier to or facilitator of its implementation or effect. Thus, it is essential to understand the context–including identifying which contextual factors influence a particular quality improvement intervention and how they do so [9,12]. As part of an implementation project aimed at contributing to reduced maternal and neonatal mortality and morbidity in a health zone in the South Kivu province of the DRC, this study's objective was to explore contextual factors influencing a training intervention focusing on healthcare practice during childbirth. The lessons learned from the results are presumed to also be useful in other similar contexts, in the DRC as well as low-income countries elsewhere, when designing and implementing similar training interventions.

## Method

### Study design

The study was approved by the National Ethical Committee of Public Health: CNES 001/DPSKI/129PM/2019. A qualitative research design was used [13], and data was collected through focus-group discussions (FGDs) with healthcare professionals participating in the training intervention.

### Setting

The DRC comprises 26 provinces, with more than 500 health zones which are organised to deliver healthcare at three levels. The primary level of care is offered at healthcare centres, some of which also offer perinatal care. The secondary level is offered at district hospitals, which have the capacity to perform C-sections, and the tertiary level is offered at referral hospitals (one per health zone) [14]. The healthcare facilities are governed by either the governmental or private sector.

The health zone where this implementation project took place is one of three in the provincial capital in the South Kivu province, situated in the eastern part of the DRC. At the time of the intervention, this health zone served more than 450,000 inhabitants at 40 healthcare facilities, of which 34 were healthcare centres, five were district hospitals, and one was a referral

**Table 1. Level, governance, and number of births at each healthcare facility.**

|  | Healthcare facilities | | | | | | |
|  | 1 | 2 | 3 | 4 | 5 | 6 | 7 |
| --- | --- | --- | --- | --- | --- | --- | --- |
| Level of healthcare | Tertiary | Secondary | Secondary | Primary | Secondary | Primary | Primary |
| Financial and governance structure | Private | Private | Private | Private | Private | Private | Private |
| Births in 2018 | 3,229 | 847 | 1,249 | 547 | 418 | 921 | 165 |

hospital. This study is part of a maternal and newborn healthcare quality improvement project being conducted at seven of these 40 healthcare facilities: three healthcare centres, three district hospitals, and the referral hospital. There were 16,101 registered births in the health zone in 2018, 7,416 of which occurred at these seven healthcare facilities. The facilities' level of healthcare and governance are shown in Table 1:

## Intervention

The training programme was developed based on core principles of conducting person-centred holistic care [15,16] as well as a woman-centred model of childbirth care [17], and with the overall aim to promote healthy physiologic, vaginal birth. The programme was divided into three pillars, with activities to 1) promote normal physiologic birth, 2) prevent and manage complications during labour and birth, and 3) strengthen the healthcare professionals' self-reflection skills and self-confidence.

The first and second pillars consisted of theory and simulation-based training using equipment from the Laerdal foundation [18], and additional tools such as birthing balls and Rebozo sheets. The first pillar was based on principles included in the midwifery model of woman-centred care during childbirth [17] and in the Rebozo technique [19]. The second pillar was based on the programmes Helping Mothers Survive Bleeding after Birth (HMS-BAB) developed by Jhpiego [20], and Helping Babies Breathe (HBB) developed by the American Academy of Pediatrics [21]. Pillar 3 consisted of reflection in groups based on a process-oriented group reflection model in which the participants reflect on themselves selected own experienced situations related to their professional role [22].

The implementation of the training programme was planned and steered by a multiprofessional project committee of healthcare professionals from the DRC (n = 3) and Sweden (n = 3), after having been developed by the Swedish research group. Details of the implementation are described in Table 2. The Congolese partners chose seven of the 40 healthcare facilities, representing all three healthcare levels. Each facility selected its own training facilitators. A 25-day training in the three-pillar programme was first given to four selected master trainers, of whom three were nurses working as midwives and one was a physician specialising in gynaecology. Next, the master trainers gave a six-day training to 13 selected training facilitators–two from each of six healthcare facilities, and one from the smallest one–consisting of ten nurse/midwives, two gynaecologists, and one paediatrician. Next, the seven healthcare facilities were equipped with equipment to conduct the training. This included uniquely developed didactic teaching material for using the programme, a birthing ball, a specially designed sheet for use of the Rebozo technique [19], and the Laerdal products 'MamaNatalie Complete', 'MamaBirthie', and 'NeoNatalie Complete' [18]. Further, based on the 'low-dose high-frequency' training pedagogy [21], a detailed schedule was defined for doing week-to-week short training activities for a period of six months, including weekly training in Pillars 1 and 2 and process-oriented reflections in groups once per week. The master trainers mentored the facilitators at the healthcare facilities and led the process-oriented reflections with the staff. The master trainers received a small monetary incentive, while the facilitators did not.

**Table 2. Description of the implementation of the three-pillar training programme.**

| Activities | Content | Time |
|---|---|---|
| Project planning | Development of didactic teaching materials and procurement of training equipment | 3 months |
| | Setup of steering committee | |
| | Identification of healthcare facilities (one hospital on tertiary level, three hospitals on secondary level, and three healthcare centres) | |
| | Introducing the local health zone authorities to the training programme and its concept, and obtaining approval | |
| | Introducing the participating healthcare facilities' management and staff to the three-pillar training programme and its concept | |
| | Collection of baseline statistics on labour and birth | |
| | Equipment assessments conducted at the seven facilities using a) the Jhpiego checklists, and b) the HMS/HBB training list | |
| | Pre-intervention discussions on contextual barriers and facilitators with the healthcare professionals at the participating facilities | |
| Training local master trainers | Training of four master trainers (three midwives and one physician) | 25 (14+11) days |
| Training local facilitators | Training of 13 facilitators (two from each of six facilities and one from the smallest one) | |
| Training healthcare professionals | Distribution of teaching materials, training equipment, and training schedule | 1 month |
| | Introduction of training programme by local facilitators | |
| Continuously facilitation | Monthly visits to each healthcare facility by master trainers | 6 months |
| | Weekly practice by local facilitators using the low-dose high-frequency practice | |
| Follow-up visits to each facility | Dialogue with facilitators and healthcare professionals at each healthcare facility, providing opportunities to share experiences of implementing the training programme. | Last month of the training programme |
| Follow-up meeting with the master trainers | Dialogue with master trainers, providing opportunities to share experiences of working with implementing the training programme | |

## Data collection

Data was gathered through FGDs in two periods: before the training started (FGDs = 7) and at the end of the programme, when facilitators and master trainers were also interviewed (FGDs = 9). The local project leader of the training programme (SNM) contacted the health zone authorities and the managers at each participating healthcare facility and informed them about the study. Authorities from the health zone and managers at each healthcare facility approved it, and provided contact information for available healthcare staff working at the maternity unit who had taken part in the three-pillar training programme. The project leader contacted these individuals and invited them to participate in FGDs after giving them verbal and written information about the study, including the fact that participation was voluntary and that they had the right to withdraw at any time without explanation. All invited healthcare professionals (n = 61), being either nurses, midwives, or gynaecologists, agreed to participate and signed informed consent.

All 16 FGDs were conducted by two of us authors (MBo and MBe). There were three to seven participants in each group. The discussions were led by MBe in French, based on an interview guide (see S1 Appendix), and were translated continuously during the FGDs into

**Table 3. Examples of the data analysis process from meaning unit to category.**

| Meaning Unit | Code | Subcategory | Category |
|---|---|---|---|
| We, as trained [facilitators, don't get any sugar [compensation] for the training we hold [at the healthcare facilities], and we didn't get any compensation when we took part in the training to become facilitators | Compensation is expected | Monetary incentives for participating are expected | Incentives influence participants' efforts to begin a training programme |

Swedish to MBo, who made field notes and asked clarifying questions. The FGDs were audio-recorded and lasted 30 to 60 minutes, with a mean of 45 minutes.

## Data analysis

The audio-recorded FGDs were analysed following principles of qualitative inductive conducted analysis [23]. First, all transcripts were read several times. Next, in new readings, meaning units were identified that answered the research question 'What are the contextual factors influencing the three-pillar training intervention and how do they influence it?' The meaning units were then compared and sorted into codes based on similar content, which were thereafter compared and clustered into subcategories and categories. The analysis process was completed by MBo and MBe separately, with repeated discussions until full agreement was reached. An example of the analysis process is shown in Table 3.

## Results

Contextual factors identified as influencing the implementation of the three-pillar training programme were sorted into three generic categories with respective subcategories; for an overview, see Table 4. In the presentation of the results the FGDs conducted in the two periods are labelled FGD 1 and FGD 2, respectively, with the facilities where they were held labelled 1–7 (see Table 1).

### Incentives motivate participants' efforts to begin a training programme

The incentives that influenced participants' efforts to get the three-pillar training programme up and running consist of three subcategories, as follows.

**Gaining increased knowledge and skills motivates.** Motivation was high among the participants to take part in the three-pillar training programme as it provided them with updated knowledge and skills for daily practice, which could contribute to healthy and positive childbirth.

**Table 4. Categories and subcategories describing contextual factors influencing the three-pillar training intervention.**

| Generic category | Subcategory |
|---|---|
| Incentives motivate participants' efforts to begin a training programme | Gaining increased knowledge and skills motivates |
| | Women's utilisation of the healthcare facilities motivates |
| | Monetary incentives for participating are expected |
| The importance of involving the local health authorities | Authorities from the health zone need to be involved |
| | The healthcare facilities' management needs to be involved |
| The need to have physical space, electricity, and equipment in place | Inadequate physical space and electricity |
| | Lack of equipment to promote physiologic birth |

Most of the participating healthcare professionals had not received any formal in-service training since their professional education, and the three-pillar training programme provided them with new knowledge. At two facilities they typically arranged training themselves in areas where care was not optimally conducted, for example in how to resuscitate a newborn. At one facility, the church sometimes organised training, and those who had participated shared their new knowledge with colleagues:

> *There is a lack of in-service training. We get nothing. Sometimes we're invited to seminars organised by the private health sector, but no such seminars are organised by the government health zone; they just give some random information.* (FGD 1, Healthcare Facility 7)

> *Most of us have not gotten any formal in-clinic training since we completed our pre-service professional education. We try to solve this through organising our own in-service trainings using our doctors, who have been at the university. But they only have their own notes, and would have needed to have PowerPoints and other educational materials.* (FGD 1, Healthcare Facility 4)

All participants, except for at one healthcare facility, expressed an awareness that they did not have the latest scientific evidence-based knowledge and accordingly were not practising optimally. They were primarily motivated to learn in several areas such as promoting normal physiologic birth and correctly managing complications like postpartum haemorrhage. The materials they were provided–birthing balls and Rebozo sheets for use during labour, and penguins and a ventilation mask for aspirating the newborn in need–added to their motivation and their possibility to practise the skills they had learned:

> *We don't have an ambulance that we can use to transport the women who need a C-section, and the ambulance from the reference hospital often comes much too late. The training we've gotten helps us support a normal birth and lets us handle acute conditions like bleeding.* (FGD 2, Healthcare Facility 6)

There was a positive attitude regarding sharing knowledge between the different healthcare facilities within the same health zone. The three-pillar training was regarded as such a knowledge exchange programme both within an individual healthcare facility as well as between the different facilities. There was a desire to develop such knowledge sharing even further, as the healthcare facilities had the same type of patients with similar backgrounds and health conditions. The master trainers, in turn, acknowledged that the training programme had given them a mandate to have access to and connect with the other facilities within the zone.

**Women's utilisation of the healthcare facilities motivates.**   Another motivating factor for the healthcare professionals to participate in the training programme was when they noticed that their changed care routines had influenced how women informed their peers about their positive experiences of being cared for at the healthcare facilities, which in turn positively influenced other women's decisions to seek care at the facilities. It had also been noticed that several women arrived earlier when their labour had started, which in turn influenced the outcome:

> *The training was fantastic, both for us as staff at the clinic and for the women who come in and give birth. At one of the clinics, many more women giving birth are coming in. Now, when a woman in labour comes, the staff are close to the woman and massage her. When the woman goes home she tells others where she lives about her positive experience.* (FGD 2 with master trainers)

It was acknowledged that caring for an extremely poor, and often low-educated, population was challenging. Women commonly feared having a C-section. With the newly trained care routines that promoted a normal physiologic birth, there was a belief among the healthcare professionals that their increased knowledge and skills in turn would increase the prevalence of vaginal, non-instrumental births, which in turn would motivate women to give birth at the facility:

*The population is very poor and has a low education level; this makes it difficult to motivate the patients for different decisions about care.* (FGD 1, Healthcare Facility 4)

*The pregnant women are afraid of having repeated C-sections. There are different reasons; one can be that the family force her to give birth normally in order to be considered a real woman. Another reason for rejecting a C-section is the cost; they therefore reject having a C-section.* (FGD 1, Healthcare Facility 5)

**Monetary incentives for participating are expected.** Another strong, motivating factor for participating in the training programme was the expectation of monetary incentives. This expectation was the same for master trainers, facilitators, and the healthcare professionals at the facilities, and was based on the fact that monetary incentives were commonly provided by other projects they had participated in. As this training project had no such monitoring incentive system in place for the weekly participation in training, it made participants at the healthcare facilities hesitant to attend the sessions:

*You know how it is with Africans: they don't come if they don't get any sugar [compensation]. You need something to motivate them; money's needed as motivation. // We, as trained [facilitators], don't get any sugar [compensation] for the training we hold [at the healthcare facilities], and we didn't get any compensation when we took part in the training to become facilitators.* (FGD 2 with facilitators)

The healthcare professionals at each facility followed work schedules covering 24 hours, seven days a week, which made it difficult for them to attend every scheduled training activity. Activities that were part of the Pillar 1 and 2 trainings were often practised in the morning, which made it challenging for those who had been working the night shift. The importance of being reimbursed as motivation to participate in the training was stressed, at least being reimbursed for transportation costs if the training activities were undertaken when someone was off duty:

*It can be hard to convince the staff to take part in the different training steps. What's hard is motivating the staff to stay and train after the end of their workday, as well as motivating staff to come in on their day off to train. Sometimes they refuse to come in because they're not paid for the transportation.* (FGD 2 with facilitators)

The master trainers stressed that the provision of incentives would increase their motivation to conduct the scheduled training activities. This, as being a master trainer and a facilitator was regarded as having dual work responsibilities–both their ordinary work as well as this training–which therefore required sufficient monetary incentives. The lack of monetary incentives to facilitators and the insufficient incentives to master trainers contributed to a lack of motivation:

*We'd like to get paid when we train the others; we only get a transportation allowance. When we had our master training in April and May, we didn't get any compensation.* (FGD 2 with master trainers)

*Because facilitators don't get paid, they're not motivated to train their colleagues.* (FGD 2 with master trainers)

### The importance of involving the local health authorities

This category describes the importance of involving both the authorities from the health zone as well as the health management at the participating healthcare facilities in the three-pillar training programme, in order to make it more successful and sustainable.

**Authorities from the health zone need to be included.** Getting the local health zone authority involved was critical for successful implementation, and obtaining this authority's approval was perceived as a requirement for conducting the three-pillar training programme. Involving the local health authority in the training could therefore support and encourage the healthcare facilities to include the training activities within their daily routines. The authorities from the health zone were informed about the training action and had granted permission to conduct it; but to support the project further, according to the master trainers, the health inspectors would need to be incentivised, and it was suggested that the project consider budgeting for this:

*The health inspectors from the health zone are included to some degree, but they're not motivated to support the project. They want to be a part of this project. They want to be there when we master trainers go to the clinics, but they don't want to be trainers. If they were included more they'd be able to motivate the staff to participate in the project during their regular visits to the healthcare facilities. . . . The inspectors have expressed that if they're paid, like they are in other projects, they can encourage the healthcare facilities to take part in the training.* (FGD 2 with master trainers)

**The healthcare facilities' management needs to be involved.** Involving the healthcare management at each healthcare facility was stressed to be of critical importance, as they make all the decisions concerning care and care routines. And if the training programme was to gain sustainability and continue beyond its scheduled time, creating ownership among the local management was acknowledged as crucial:

*We have a culture in which the responsible parties are higher in rank than us; we others feel lower than them. So if this training is to continue, you have to involve more of those at management level at the healthcare facilities. That would increase the ownership. Then, management will take greater responsibility, they'll increase our motivation, create an ownership.* (FGD 2, Healthcare Facility 1)

Being involved entailed not only being informed about and influencing the training strategy, which was a part of the project; it also included receiving monetary incentives. If not, this could act as an obstacle to the training programme. This was especially clear at the tertiary-level facility. These leaders hindered the programme in various ways, for example by not taking part in the training and even stating that they had never heard about it:

*If people with management responsibility aren't directly involved in the project, they'll turn the responsibility over to those who are running the project. You have to bring in the boss, make him part of the project. When the boss talks everybody listens; I can't ask the boss to do*

*various things, I can't give the boss orders. So the boss has to participate and have a mandate. And that means that the boss needs money from the project. It's not a salary; it's a motivation. If the boss is motivated, we can get everybody to do what we want in the project. If he doesn't get paid he's not going to participate himself and he's going to work against it. The other healthcare facilities are small; this is a gigantic clinic, and the boss has to be involved.* (FGD 2 with master trainers)

## The need to have physical space, electricity, and equipment in place

This category describes, in two subcategories, the need to have physical space, electricity, and equipment in place in order to carry out the three-pillar training programme.

**Inadequate physical space and electricity.** According to the participants and our own observations all seven facilities, and specifically their maternity units, had inadequate physical space to meet the needs. Hence, this contradicted the training. One challenge that negatively influenced the training using mannequins was that the mannequins were packed in bags and stored, and were only picked up for each training session. According to the participants this was due to a lack of space, and the fact that there was no separate table for the mannequins. This resulted in the training not always being conducted as they had learned during their training. Another reason for storing the training products was a fear that the material would be stolen:

> *At the labour ward there's no place; we've notified the staff that they can use the material on the day. We have no room to store it openly and guard it. We have many interns, the African culture–the material can be stolen.* (FGD 2 with master trainers)

No access to, or insufficient availability of, electricity was another challenge that limited the possibilities to conduct care during labour and birth based on the staff's new knowledge. Often, electricity was only available for six to eight hours or even less, but there could also be two days with no electricity at all. This led to ambiguity about using electric equipment, as it was almost impossible to rely on equipment like the blood refrigerator, resuscitation equipment, and heating lamps for newborns, which depended on electricity. Solar panels were commonly used, but were often insufficient. At one facility, the blood refrigerator used all the electric capacity. Thus, it was motivating to take part in this training programme and use the assigned equipment that did not require electricity:

> *Electricity comes and goes; we cannot say how often. We have solar cells but they're not strong enough to drive medical equipment, as it needs electricity.* (FGD 1, Healthcare Facility 6)

**Lack of equipment to promote physiologic birth.** Unanimously, all participants mentioned the lack of childbirth care equipment at the healthcare facilities as very limiting to the provision of high-quality maternal and newborn healthcare. There was especially a lack of equipment for promoting normal physiologic birth. Thus, the birthing balls and a specially designed sheet for the use of Rebozo techniques were specifically valued, as this made it possible to offer alternative pain relief and positions during labour and birth.

The material provided through the project was used often, and all participants stressed a need for additional birthing balls and Rebozo sheets. At some facilities, the balls and sheets had been used so often they were now broken/torn, while other facilities were in need of extra equipment so that it could be used if several births were taking place at the same time. The importance of cleaning the equipment between women also created a need for extra balls and sheets:

*I participated in a one-week course last autumn and learned about alternative pain relief such as Rebozo and birthing balls, but couldn't practise it as we didn't a ball like that.* (FGD 1, Healthcare Facility 2)

*If we have two women at the same time we can't offer both women a birthing ball. We want to have a ball for all women giving birth; we have about three deliveries a day. We also need more Rebozo sheets. We'd like to have three Rebozos.* (FGD 2, Healthcare Facility 4)

## Discussion

The study identified three contextual factors which influenced the three-pillar training intervention programme aimed at improving healthcare practice during labour and birth: (i) Incentives motivated participants' efforts to begin the training programme; (ii) Involving the local health authorities was important; and (iii) Having physical space, electricity, and equipment in place was crucial. A central feature in these identified factors is the implication of incentives, which we will discuss further below.

The participants expected to be monetarily reimbursed as an add-on to their monthly salary. They stressed that participation in training and projects of different kinds mostly implies being paid, and thus expected to be incentivised in our training programme as well. Only the master trainers were reimbursed, and they judged the compensation level to be too low. This may be explained by the fact, found in another study in the DRC, that healthcare professionals in the DRC, especially nurses and midwives, often lack regular payment or compensation for their employment [24].

Barriers related to not receiving monetary incentives are not unique in quality improvement interventions. The issue has been identified in other low- and middle-income country healthcare projects in which community health workers have been used to increase the possibility to achieve the goals; there, payment positively influenced the health workers' motivation to contribute [25,26]. The possibility of monetary incentives to increase professionals' willingness to participate was also observed in a training intervention as part of a neonatal health project in Vietnam [27].

A main finding in our study was that the local health authorities also expected to be paid if they were to facilitate and encourage the training programme, even though they were not involved as trainers. In a recent Cochrane review, the involvement of local leaders in quality improvement activities was found to be effective in implementing evidence-based practice. The report stresses the importance of engaging healthcare leaders in interventions aimed at improved health outcomes [28]. Similar findings have been reported in a review article about middle managers' role in healthcare evidence-based practice implementation [29]. Our training programme was introduced and accepted by the local authorities both in the health zone and at the involved healthcare facilities, but without their being paid. At one of the participating facilities it was clearly observed that this led to a hindrance of the programme's activities and effects. As a consequence, this barrier negatively affected the expected improved maternal and newborn health outcomes. These findings are rather disappointing.

The issue of providing or not providing performance-based financing, and its effect on ensuring the delivery of high-quality health services, has been studied in the DRC, where almost 500 health workers representing five of the 26 provinces participated. Workers who had received monetary incentives which had then been stopped when the project ended scored significantly lower on most dimensions of motivation than did those who had never received money. The study highlights the potentially negative effect on health workers' motivation when large donor-driven projects provide generous incentives for participation in training,

meetings, and workshops which are then withdrawn after the project-based activities are finalised [30].

That large donors' provision of performance-based financing to health workers and leaders also influenced our training intervention was obvious, as this had created 'norms' among healthcare providers and leaders that such interventions should be paid for and, if not paid, they would not participate or could even create a barrier. Thus academic research-based projects, which usually have smaller budgets to work with, cannot compete with large donor organisations.

Another, highly positive, contextual factor and effect of our three-pillar training programme was that the participants–both the master trainers, facilitators, and healthcare professionals–recognised its value for increasing knowledge. A strong preference for learning evidence-based knowledge and skills was shown among the participants; not only to have refresher trainings, but also to learn new things–which enabled them to better conduct high-quality care during labour and birth. That continuous education and training serve as important motivation for healthcare professionals has been observed elsewhere [31]. Meanwhile, non-monetary incentives, in terms of level of decision-making among community workers in low- and middle-income countries, have been shown to be highly effective in increasing intrinsic motivation [32].

Another positive effect of the training programme was that it was immediately transferred to care practice. Positive changes in caring, such as being closer and using alternative methods such as birthing ball, had been experienced by the women who in turn had informed their peers who then wanted to give birth at the same facility.

This is an example that through low-dose high frequency training it is possible to immediately implement a more woman-centred respectful intrapartum care [17,33,34], and which is instantly told to society by women being cared.

When it comes to successfully implementing the use of HBS and HMS training programmes, which constituted pillar two of our three-pillar training programme, it has been concluded that a successful implementation requires country-led commitment, readiness, and follow-up to create local accountability and ownership [35]. Unfortunately, our study did not fully involve the health zone authority, which resulted in a lack of ownership. These findings in our study have offered novel insight regarding contextual factors of incentivising authorities if a training intervention aims to have their full involvement. Comparing the findings with those of other studies on maternal and neonatal health improvement in a low-income setting confirms the need to account for involving local authorities from various levels of the health system from the planning phase, through budgeting, and throughout the implementation and evaluation processes [27,36]. This strategy of involvement may reduce the risk of facing hindrance in terms of monetary incentives for evidence-based interventions, which are proven to have an impact on the health outcomes of mothers and newborns.

Another motivating incentive of the programme was that it provided the healthcare facilities with material–both mannequins for training purposes as well as items to use during labour and birth, such as birthing balls and a specially designed sheet for the use of Rebozo techniques. The use of these materials in combination with better humanised behaviour towards the women, which was stressed in Pillar 1, gave the healthcare facilities a more positive reputation through women's sharing of positive information with peers, and also seemed to reduce the negative trend of pregnant women's delay in going to the facilities.

## Methodological considerations

Our study is among the first to show evidence of the influence of contextual factors in a training intervention aimed at improving intrapartum care in the DRC. However, the study has its

limitations. It was carried out at only seven of 40 healthcare facilities, and in only one health zone in the DRC; this was due to limited resources. It cannot be assumed that the contextual factors influencing such an intervention in the other 33 facilities in the chosen health zone, as well as elsewhere in the DRC, would be the same. Therefore, the results may not be generalised to the entire health zone, nor the country. Another limitation is that only private healthcare facilities, and no governmental ones, were included. This may have caused us to miss other contextual factors influencing such a training intervention.

A strength of the study is the interdisciplinary mix of the participating researchers. M Bogren and M Berg, from Sweden, both hold the degrees of PhD, RM, and RN, are conducting research in low-income settings including the DRC, and have extensive experience working in low-income countries through multilateral organisations (M Bogren) and private health systems, including in the DRC (M Berg). The third author, S N. Mwambali, is a Congolese gynaecologist, and was the local project leader of the training intervention.

## Conclusions

To conclude, this study found what also has been found earlier, that aspects of the context influence the implementation of an intervention and its outcomes, and hence its feasibility and usefulness [12]. That incentives are a critical element of successful health interventions in impacting sexual, reproductive, maternal, and newborn healthcare quality in low- and middle-income countries is confirmed in a recent systematic review [37]. A critical lesson learned from this study in the DRC is that incentives of various aspects are crucial contextual factors to consider when planning for a training intervention. In a small research project like ours, fully meeting the expectations of monetary incentives would require a reduction of the scale and thus limit the implementation of new evidence-based knowledge into practice.

## Supporting information

**S1 Appendix.**
(DOCX)

## Acknowledgments

We would like to express our sincere appreciation to all the healthcare providers who participated in this study. We also want to thank Dr. Prof. Denis Mukwege, Panzi Hospital, Susheela M Engelbrecht at Jhpiego, Maria Hogenäs, Art of Life, and Marthe Byamungu Makundane for their respective contribution in the training programme.

## Author Contributions

**Conceptualization:** Malin Bogren, Marie Berg.

**Data curation:** Malin Bogren, Marie Berg.

**Formal analysis:** Malin Bogren, Marie Berg.

**Funding acquisition:** Malin Bogren, Marie Berg.

**Investigation:** Malin Bogren, Marie Berg.

**Methodology:** Malin Bogren, Marie Berg.

**Project administration:** Malin Bogren, Sylvie Nabintu Mwambali.

**Supervision:** Malin Bogren, Marie Berg.

**Validation:** Malin Bogren, Marie Berg.

**Writing – original draft:** Malin Bogren, Marie Berg.

**Writing – review & editing:** Malin Bogren, Sylvie Nabintu Mwambali, Marie Berg.

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
