## [Decision Letter · Decision Letter 0]

13 Aug 2021

PONE-D-21-15167

Contextual factors influencing a training intervention aimed at improved maternal and newborn healthcare in a health zone of the Democratic Republic of Congo

PLOS ONE

Dear Dr. Bogren,

Thank you for submitting your manuscript to PLOS ONE. After careful consideration, we feel that it has merit but does not fully meet PLOS ONE’s publication criteria as it currently stands. Therefore, we invite you to submit a revised version of the manuscript that addresses the points raised during the review process.

We look forward to receiving your revised manuscript.

Kind regards,

Ashraful (Neeloy) Alam, PhD, MSS

Academic Editor

PLOS ONE

Journal Requirements:

2. During our internal checks, the in-house editorial staff noted that you conducted research or obtained samples in another country. Please check the relevant national regulations and laws applying to foreign researchers and state whether you obtained the required permits and approvals. Please address this in your ethics statement in both the manuscript and submission information. In addition, please ensure that you have suitably acknowledged the contributions of any local collaborators involved in this work in your authorship list and/or Acknowledgements. Authorship criteria is based on the International Committee of Medical Journal Editors (ICMJE) Uniform Requirements for Manuscripts Submitted to Biomedical Journals - for further information please see here: https://journals.plos.org/plosone/s/authorship.

Reviewers' comments:

Review Comments to the Author

Reviewer #1: This paper reports on a training programme in MNH in DRC. It is an interesting read but in itself there is nothing new about evaluating the impact of a training programme, as we know that teaching/training works. The authors instead have looked at incentives for training.

MMR figures are given for 2014 but there is data available from WHO for 2017, could this be used instead

The methods section is sound and clearly written

In the results section 3 generic categories are given and the tables provided give clarity to how these were arrived at.

Category one relates to financial incentives being effective, despite what the authors say that this is relatively new evidence, it has been reported by WHO as being effective as far back as 2010. There could be more discussion on the relative merits of financial incentives versus professional incentives (that is knowledge acquisition for doing a better job). This would bring in the 3rd pillar of the training more (strengthen the healthcare professionals’ self-reflection skills and self-confidence).

There needs to be more theoretical discussion around why women access improved services (for example issues of trust)

There is nothing really new in the idea that this type of training programme needs government and local managerial support.

The authors mention the need for adequate physical space, electricity etc but then in the results mention this as part of service delivery and not just training needs. This needs to be a separate category which could emphasise more the difficulty of implementing training without an enabling environment.

The article does seem to concentrate on financial rather than non financial incentives and I wonder to make it stand out more if the real emphasis should be on non financial. This may yield a more original discussion and also link into to why if 80% of women in DRC do have a hospital assisted birth that the MMR ratio is still so high. perhpas this relates to professional attitude in practice and that could be interesting to read. it may also be due to poor facilities and equipment and some discussion of this would be good.

Reviewer #2: it is a well written manuscript. the subject matter is relevant and important the importance of contextual characteristics required to design and execute a successful training for a major impacts well highlighted in a era where significant resources are being channeled to competency training in an effort to improve QOC ,this manuscript is very useful

6. PLOS authors have the option to publish the peer review history of their article (what does this mean?). If published, this will include your full peer review and any attached files.

Reviewer #1: No

Reviewer #2: No

---

## [Editor Report · Decision Letter 1]

4 Nov 2021

Contextual factors influencing a training intervention aimed at improved maternal and newborn healthcare in a health zone of the Democratic Republic of Congo

PONE-D-21-15167R1

Dear Dr. Bogren,

We’re pleased to inform you that your manuscript has been judged scientifically suitable for publication and will be formally accepted for publication once it meets all outstanding technical requirements.

Kind regards,

Ashraful (Neeloy) Alam, PhD, MSS

Academic Editor

PLOS ONE

Additional Editor Comments (optional):

I suggest please read the manuscript carefully again to fix any typos. For example, replace 'on' with 'one' in the the 2nd sentence in the conclusion of the abstract. There might be some more such errors that need to be corrected.
---

## [Editor Report · Acceptance letter]

16 Nov 2021

PONE-D-21-15167R1 

Contextual factors influencing a training intervention aimed at improved maternal and newborn healthcare in a health zone of the Democratic Republic of Congo 

Dear Dr. Bogren:

I'm pleased to inform you that your manuscript has been deemed suitable for publication in PLOS ONE. Congratulations! Your manuscript is now with our production department. 

Kind regards, 

on behalf of

Dr. Ashraful (Neeloy) Alam 

Academic Editor

PLOS ONE